# European Patent in Immunoncology: From Immunological Principles of Implantation to Cancer Treatment

**DOI:** 10.3390/ijms20081830

**Published:** 2019-04-12

**Authors:** Franziska M. Würfel, Christoph Winterhalter, Peter Trenkwalder, Ralph M. Wirtz, Wolfgang Würfel

**Affiliations:** 1STRATIFYER Molecular Pathology GmbH, D-50935 Cologne, Werthmannstrasse 1c, 50935 Cologne, Germany; ralph.wirtz@stratifyer.de; 2INTELLEXON GmbH, Keltenstrasse 27, 82343 Pöcking, Germany; christoph.winterhalter@web.de (C.W.); peter.trenkwalder@t-online.de (P.T.); prof.w.wuerfel@ivf-muenchen.de (W.W.)

**Keywords:** cancer, granted European patent, HLA profiling, HLA-E, HLA-F, HLA-G, immune evasion, individualized antibody therapy

## Abstract

The granted European patent EP 2 561 890 describes a procedure for an immunological treatment of cancer. It is based on the principles of the HLA-supported communication of implantation and pregnancy. These principles ensure that the embryo is not rejected by the mother. In pregnancy, the placenta, more specifically the trophoblast, creates an “interface” between the embryo/fetus and the maternal immune system. Trophoblasts do not express the “original” HLA identification of the embryo/fetus (HLA-A to -DQ), but instead show the non-classical HLA groups E, F, and G. During interaction with specific receptors of NK cells (e.g., killer-immunoglobulin-like receptors (KIR)) and lymphocytes (lymphocyte-immunoglobulin-like receptors (LIL-R)), the non-classical HLA groups inhibit these immunocompetent cells outside pregnancy. However, tumors are known to be able to express these non-classical HLA groups and thus make use of an immuno-communication as in pregnancies. If this occurs, the prognosis usually worsens. This patent describes, in a first step, the profiling of the non-classical HLA groups in primary tumor tissue as well as metastases and recurrent tumors. The second step comprises tailored antibody therapies, which is the subject of this patent. In this review, we analyze the underlying mechanisms and describe the currently known differences between HLA-supported communication of implantation and that of tumors.

## 1. Introduction

The close similarities between embryos, i.e., trophoblast invasion and the invasive behavior of tumors have long been known and are the subject of many publications, observations, and hypotheses [1,2]. An editorial on this subject has been published recently [3].

The HLA system is the central focus of cellular and immunological communication; it plays an essential role in the immunological identification of cells and in communication with and within the immune system. Profiling of the HLA system is an essential diagnostic step prior to organ or bone marrow transplantations. In transplantation, there is always a direct contact between the donor organ, typified by HLA classes I and II (*HLA A-C*; *HLA DP-DS*), and the recipient’s immune system. This results in more or less severe reactions of the recipient’s immune system. Because of this, prior to transplantations, HLA matching is performed. However, even in cases of a high HLA concordance, additional immunosuppressive therapy is usually necessary.

However, conditions are different in implantation and pregnancy. The embryo/fetus itself does not come into direct contact with the maternal immune system, but the trophoblast and/or placenta does. Although the trophoblastic cells are of embryonic origin, they show a different HLA identification compared to adult epithelial tissue: no classical HLA class I and II proteins (*HLA-A*, *HLA-B*, and *HLA DP-DS*) from the original embryonic/fetal HLA signature are expressed, except for HLA-C [4,5,6,7]. Instead, the “non-classical” HLA groups (class Ib) such as *HLA-E* [8,9,10], *HLA-F* [10,11], and *HLA-G* [12,13,14,15,16,17,18] are expressed. Despite the indirect cell contact of the maternal and fetal classical HLA system, the fetus has a minimum of 50% non-matching HLA compared to the mother. The 50% HLA “miss-match” is derived from the future father; therefore, the embryo is often described as a “semi-allotransplant”. Moreover, in pregnancies after egg or embryo donation or in cases of surrogacy, the embryo is, genetically, completely foreign to the future mother, which might be regarded as “allotransplant”. However, these pregnancies are also successful, even though the mother is not related to the embryo. In these cases, as well as standard mother-to-embryo constellations, implantation and pregnancy comprise a close cellular “network” of the trophoblast and maternal tissue. Due to this special situation, the trophoblast, to maintain pregnancy, expresses the “non-classical” HLA groups in order to escape the maternal immune system. The “non-classical” HLA groups inhibit immunocompetent cells of the maternal immune system by interacting with specific receptors of NK cells (e.g., killer-immunoglobulin-like receptors (KIR)) and lymphocytes (lymphocyte-immunoglobulin-like receptors (LIL-R)). In pregnancy failures and placental disorders, because of an aberrant expression of the non-classic HLA groups, this mechanism is not working properly. Tumor cells use the same immune escape mechanism to evade immune cell elimination. In view of this context, it is necessary to examine these non-classical groups and their interaction with the receptors/receptor families of the various immunocompetent cells more closely.

## 2. Non-Classical HLA Groups E to G (class Ib)

### 2.1. HLA-E

*HLA-E* is the first prominent non-classical HLA class Ib members. It is located 650 kb upstream from *HLA-C* and flanked by *HLA-A* on the 3’ position on chromosome 6p21.3. Similar to *HLA-B*, it consists of seven exons. The first exon encodes the signal peptide, while exons 2, 3, and 4 encode the alpha domains α1, α2, and α3. Exon 5 encodes the transmembrane domain, and exons 6 and 7 encode the cytoplasmatic part of the protein [19].

HLA-E is the least polymorphous HLA antigen with 13 known alleles, resulting in five different proteins [20,21]. The alleles *HLA-E*0101* and *HLA-E*0103* are the two known functional alleles. They differ by an amino acid substitution (arginine for glycine) at position 107, which results in a differentiated cell surface expression, thermal stability, and peptide binding affinity [22,23]. For a stable surface expression, HLA-E needs to bind intracellular peptide sequences, exclusively derived from signal peptides of classical HLA class I proteins and HLA-G [20,24,25]. Surface expression of HLA-E therefore also depends on expression of the classical HLA groups (class Ia) and HLA-G. Reduced expression of classic HLA groups and HLA-G thus results in a reduced HLA-E expression [23,26]. Besides signal peptide sequences from other HLA groups, HLA-E also binds peptides derived from cell-stress-related proteins such as Hsp60 and pathogen-associated proteins such as the human cytomegalovirus [27,28]. In contrast to other non-classical HLA class I genes, HLA-E is not only expressed in trophoblast cells but also in a variety of nucleated cells [29]. In the trophoblast, HLA-E is mainly expressed during the first trimester of pregnancy [30].

### 2.2. HLA-F

*HLA-F* is the second but lesser known member of the non-classical HLA class I family. *HLA-F* is located on the terminal end of chromosome 6p21.3. Similar to the HLA class Ia genes, the *HLA-F* gene has eight exons, which likewise encode the signal peptide and the alpha and transmembrane domains. The functions of these exons are similar to *HLA-E*: the first exon encodes the signal peptide, while exons 2, 3, and 4 encode the alpha domains α1, α2, and α3. Exon 5 encodes the transmembrane domain, while exons 6 and 7 encode the cytoplasmatic part [19]. Thus far, 22 *HLA-F* alleles with three mRNA transcript variants—*HLA-F1*, *-F2*, and -*F3*—are currently known [21]. However, recent NGS analysis determined 37 haplotypes, resulting in 30 coding haplotypes on the *HLA-F* region [31]. Unlike the classical HLA groups, exons 7 and 8 are not translated in the transcript variants F1 and F3 [19]. The cytoplasmatic tail is shorter, compared to other HLA genes, and shows greater variation in length between the three isoforms [32]. However, the structure of the protein corresponds to the classic HLA class Ia groups and forms a peptide binding pocket using the α1 and α2 domains. In this classical conformation, HLA-F forms a complex with β2-microglobulin. Besides its ability to bind to β-microglobulin, HLA-F also forms complexes with heavy chains of other HLA class I molecules, possibly to stabilize them [33]. Apart from the classical HLA conformation and complexes with other heavy HLA chains, there is a stable open conformation (OC) of HLA-F characterized by the absence of β2-microglobulin and peptides bound in the peptide binding groove [32].

HLA-F expression is mainly restricted to the intracellular parts of the cell, specifically the endoplasmatic reticulum, but can also be expressed on the surface of trophoblastic and activated cells of lymphocytic origin, e.g., activated B-cells [34]. High expression is restricted to the extravillous trophoblast and declining in the further course of pregnancy [30].

### 2.3. HLA-G

*HLA-G* is the most prominent gene within the non-classical HLA group. Similar to other HLA genes, it is located on chromosome 6p21.3: flanked upstream by *HLA-A* and downstream by *HLA-F*. *HLA-G* remains the most polymorphic gene within the otherwise almost non-polymorphic non-classical HLA group. At present, 75 single nucleotide polymorphisms (SNPs) are known for the coding region of *HLA-G*, resulting in 53 alleles with 18 proteins and associated isoforms [21,35,36]. Similar to the HLA class Ia groups, *HLA-G* has 8 exons, but the numbering of exons differs between databases. The National Center for Biotechnology Information (NCBI) and the genomic browser from the University of California Santa Cruz (UCSC) start with exon 1 in the possible promoter sequence of *HLA-G*, resulting in a translational start for the signal peptide in exon 2. Exons 3–5 encode the alpha 1–3 domains, and exon 6 encodes the connecting peptide, the transmembrane domain, and the cytoplasmatic tail. The IMGT/HLA database denotes the exon that encodes the signal peptide as exon 1, resulting in a shifted exon numbering for the translated protein. In all databases, exon 8 remains untranslated due to the presence of a stop code in exon 7. The region of the non-translated exon 8 is designated as the “3´untranslated region (3′UTR)” [37]. Eighteen SNPs, a 14 bp insertion/deletion, and 44 haplotypes are currently described for the 3′UTR region [36], which are known to influence the translation of HLA-G proteins by reduced transcription, mRNA stability, or aberrant alternative splicing. Besides sequence variants, six micro-RNAs (miRNAs) (miR-133a, miR-148a, miR-148b, miR-152, miR-548q, and miR628-5q) are also known to downregulate HLA-G protein expression by binding to the 3′UTR. [36].

Due to alternative splicing of the primary transcript, HLA-G can be expressed as four membrane-bound isoforms (HLA-G1 to -G4) and, in contrast to classical HLA class I genes, as three soluble isoforms (HLA-G5 to -G7) [38]. HLA-G1 and the soluble HLA-G5 represent the complete extracellular protein structure composed of three alpha domains (α_1_–α_3_) and may be bound to β2-microglobulin (β2m) [39]. The other isoforms are not associated with β2m and differ in their extracellular protein structure. The soluble isoforms show a high structural similarity with the membrane bound isoforms HLA-G1 and HLA-G2 but preserve the intron 4 (HLA-G5 and -G6) [40,41,42]. HLA-G7 is the soluble structural homolog of HLA-G3 and comprises only the α1 domain. The absence of transmembrane domains is the result of the translation of intron 2, which encodes two amino acids bound to the α1 domain [37]. Besides the different known protein isoforms, proteasome-generated spliced peptides for classical HLA groups have been described, accounting for approximately one-third of the whole antigens [43]. Considering the genetic and structural similarities between classical HLA class I genes and HLA-G, further HLA-G peptides and proteins could be generated through this post-transcriptional modification. It is assumed that they have an impact on T-cell response [44].

In addition to these isoforms, HLA-G can also occur in several protein complex formations. Similar to the classic HLA proteins, HLA-G1 exists as a monomer bound with β2-microglobulin. HLA-G1 and HLA-G5 are able to form heterodimers and homotrimers by establishing disulfide bridges between the α1 domains (via cysteine 42). The remaining membranous isoforms do not form a complex with β2-microglobulin, but, similar to HLA-G1 and HLA-G5, can also form hetero- and homodimers.

Compared to HLA-E and -F, HLA-G protein expression is only restricted to trophoblast cells in pregnancy. HLA-G1 is the main isoform and is expressed in extravillous trophoblast cells (EVTs), e.g., endothelial EVTs, interstitial EVTs, and villous trophoblast cells. EVTs further express the isoforms HLA-G2, -G5, and -G6, while the villous trophoblast cells only express the soluble homolog of HLA-G1, i.e., HLA-G5, which is released into the maternal bloodstream [13,45].

## 3. Interaction of HLA-E to -G with Receptor Families of Immunocompetent Cells

As mentioned above, the non-classical HLA genes predominantly mediate immune evasion and immune suppression by inhibiting cells of the adaptive and innate immune system, e.g., natural killer cells (NK cells), T- and B-lymphocytes by interacting with the inhibitory leukocyte immunoglobulin-like receptors B1 and B2 (LILRB1 and LILRB2), killer cell immunoglobulin-like receptor 2DL4 (KIR2DL4), and the NK receptor group 2 (NKG2)/killer cell lecithin-like receptors (KLRs) [46,47].

### 3.1. Receptor Interaction of HLA-E

HLA-E interacts with the killer cell lectin-like receptor C1 (KLRC1), also known as NKG2A, -B and -C which is expressed by NK cells [48]. The NKG2A receptor belongs to the NKG2 receptor family, which represents the second group of NK cell receptors, beside KI-receptors (KIR) [8,49]. These receptors are type II transmembrane proteins that have an extracellular C-type transmembrane domain. The NKG2 receptor family can be subdivided into six subgroups: -A, -B, -C, -D, -E, and -H. Some are splicing variants of the same gene (A/B and E/H) [49]. The NKG2A receptor together with the NKG2B receptor belongs to the inhibitory receptors within the NKG2 family, which mediate an inhibitory signal to the NK cell via immunoreceptor tyrosine-based inhibition motifs (ITIMs) [50]. Upon binding of HLA-E to NKG2A, the ITIMs bind to the Src homology region 2 domain-containing phosphatase-1 (SHP-1), which dephosphorylates signaling molecules in the signaling pathway of the immunoreceptor tyrosine-based activating motifs (ITAMs) and thus sends an inhibitory signal to the cell [51]. Similar to NKG2B, -C, -E, and -H, the NKG2A receptor dimerizes with CD94 to form a stable complex [52,53]. CD94 is expressed by NK cells and plays a role in recognizing HLA class I proteins. With the exception of CD94/NKG2D, the CD94/NKG2 receptor family only recognizes HLA-E as a ligand [52]. However, the affinity of HLA-E towards single NKG2 subtypes varies. HLA-E binds preferentially to the inhibitory receptor CD94/NKG2A rather than the activating receptor CD94/NKG2C (Figure 1) [52]. Activating receptors such as CD94/NKG2C do not possess intracellular motifs. For signaling transduction, the CD94/NKG2C receptor contains a positively charged transmembrane domain with DNAX activation protein 12 (DAP-12), which has an ITAM in its cytoplasmic domain and transmits an activating signal to the cell [51,54,55]. Once activated, the signaling transduction protein DAP12 recruits the spleen tyrosine kinase (Syk) and zeta-chain-associated protein kinase 70 (ZAP70), which stimulate increased cytotoxicity and cytokine production in NK cells [56]. HLA-E inhibits NK cell activation and proliferation via the inhibitory NKG2A receptor [57]. Activation via the NKG2C receptor only occurs after binding of a restricted repertoire of peptides such as CMV, A80 and B13 [58]. However, HLA-E that has bound HLA-G leader peptide sequence engages, the activating receptor of which is CD94/NKG2C. This binding results in an increased proliferative activity and elevated antibody-dependent cytotoxicity [29].

### 3.2. Receptor Interaction of HLA-F

HLA-F interacts mainly with the receptors LILRB1 and LILRB2 as a tetramer belonging to the family of leukocyte-immunoglobulin (Ig)-like receptors (LILR) (Figure 2) [34,59]. They comprise five activating (A) receptors (LIRA1, -2, -4, -5 and -6), one soluble receptor (LIRA3), and five inhibitory (B; “blocking”) receptors (LILRB1–5) [60]. The LILRB1 receptor (also known as Ig-like transcript 2 (ILT2) or CD85j) is an inhibitory receptor expressed in monocytes, dendritic cells (DCs), as well as B-, T-, and NK cells. The receptor has four ITIMs in its intracellular domain and transmits an inhibitory signal to the immunocompetent cell via SHP-1 and -2. During T-cell activation, cytotoxicity is inhibited exclusively via SHP-2, which induces blockade of the mechanistic target of rapamycin (mTOR) signaling pathway [61]. The LILRB1 receptor is able to prevent the proliferation of antigen-specific γδ-T-cells by forming complexes with further LILRB1 receptors [62]. This homodimer of LILRB1 receptors induces the synthesis of interleukin-10 (IL-10) and the transformation of growth factor beta (TGF-β) and inhibits the synthesis of the proinflammatory cytokine interferon-gamma (IFN-y) [46,63,64]. The LILRB2 receptor, also known as ILT4, belongs to the inhibitory receptors and is primarily expressed by dendritic and endothelial cells [65]. Unlike LILRB1, LILRB2 has only three ITIMs, which are able to recruit SHP-1 and -2, mediating an inhibitory signal to the DCs [39]. The receptors LILRB1 and B2 bind to β2-microglobulin and to the α3 domain of the MHC Class I and recognize numerous HLA Ia and Ib molecules in addition to HLA-F.

HLA-F as an open conformer (OC) is able to bind to the activating NK cell receptor KIR3DS1, KIR2DS4 and to the inhibitory NK cell receptor KIR3DL1 and -2 [66,67,68]. The receptors KIR3DS1, KIR2DS4, KIR3DL1 and -2 belong to the family of killer cell immunoglobulin-like receptors (KIRs), which are expressed in NK cells as transmembrane glycoproteins. They are distinguished and classified by their number of extracellular domains (two or three domains, 2D or 3D) and the length of their intracellular signaling domains. KIR subtypes with short (S) intracellular domains are classified as activating receptors because their transmembrane domain contains a charged lysine residue instead of an immunoreceptor tyrosine-based inhibition motif (ITIM) [69]. KIRs such as KIR3DL1 with a long cytoplasmic domain (L) belong to the category of inhibitory receptors; their cytoplasmic domain also contains an ITIM, which mediates an inhibitory signal to NK cells after activation by the protein tyrosine phosphatases SHP-1 and SHP-2 [70,71]. However, HLA-F OC possesses the highest binding affinity for the activating receptor KIR3DS1. Via the charged transmembrane domain, KIR3DS1 dimerizes with DAP-12, which transmits an activating signal to the NK cell via ITAMs. Binding of HLA-F to KIR3DS1 results in NK cell activation [72,73].

### 3.3. Receptor Interaction of HLA-G

HLA-G interacts with various receptors, such as LILRB1, LILRB2, CD8, CD160 and KIR2DL4, which are expressed by cells of the adaptive and natural immune system (Figure 3) [36]. The inhibitory receptor LILRB1 is found in monocytes, DCs, and B-, T-, and NK cells: The LILRB2 receptor is only expressed in dendritic and on endothelial cells, which bind to β2-microglobulin and to the α3 domain and thus recognize numerous HLA class Ia and Ib molecules. Because of the hydrophobicity of the α3 domain in the HLA-G molecule, receptors favor HLA-G as an interaction partner for binding [74]. The binding affinity of receptors LILRB1 and LILRB2 increases upon multimerization of the HLA-G molecules [38]. However, the two receptors recognize different structure conformers of HLA-G: LILRB1 preferentially binds to HLA-G in a complex with β2-microglobulin, while LILRB2 also recognizes the free heavy alpha domains of HLA-G without bound β2-microglobulin [75,76]. The interaction of soluble and membrane-bound HLA-G with LILRB1 or LILRB2 causes direct and indirect immune-suppressive effects on immune cells [77]. The direct immune inhibitory mechanism is mediated by the direct binding of HLA-G to these receptors, which causes an inhibition of cytotoxic T-cells as well as NK cells [78,79]. As an indirect effect HLA-G inhibits the proliferation of allo-specific CD+ T-cells [80]. As the LILRB1 receptor is also expressed by DCs and B-cells, the interaction with HLA-G affects their function and maturation as well and induces the generation of HLA-G expressing and tolerogenic DCs such as DC-10 [81]. HLA-G+ APCs can induce immunosuppressive CD4+ T cells and, in the case of DC-10, mediate the generation of type 1 regulatory T cells [82,83]. The receptor KIR2DL4 (also known as CD158d) is a special type among KIR receptor subtypes. Unlike other KIR receptors, KIR2DL4 interacts within the non-classical HLA group only with HLA-G [55]. The receptor contains the two extracellular domains D0 and D1 and has only one ITIM. A charged arginine residue enables KIR2DL4 to form a complex with Fc fragment receptor γ (FcRγ). FcRγ has two ITAMs, which transmit the activation signal to the NK cells. The complex formed with FcRγ stimulates the NK cell to enhanced chemokine and cytokine production [55]. Unlike other KIR receptors, KIR2DL4 cannot be detected on quiescent peripheral NK cells with phenotype CD56^dim^CD16+. It is primarily expressed in an NK cell subpopulation known as decidual/uterine NK cells (uNK) with the phenotype CD56^bright^CD16- [84,85,86]. The membrane-bound HLA-G binds to KIR2DL4, stimulating the uNK cells to proliferate and produce IFN-γ [87].

HLA-G also interacts with CD8, a surface marker for cytotoxic T-cells. Binding of soluble HLA-G (sHLA-G) to CD8 induces apoptosis in CD8+ cytotoxic T-cells [88]. In addition to cytotoxic T-cells, CD8 is also expressed in some activated uNK cells [90]. Engagement of HLA-G with CD8 expressed on uNK cells likewise induces apoptosis in this subpopulation [91].

Besides uNK and cytotoxic T-cells, sHLA-G also induces apoptosis in activated endothelial cells through the engagement with the CD160 receptor [92]. CD160 is expressed on CD56dim CD16+ cytotoxic NK cells, CD8+ T lymphocytes, and activated endothelial cells [92]. It is a glycosylphosphatidylinositol-anchored receptor that cannot mediate signals by itself due to missing ITAMs. The receptor relies on the recruitment of adapter proteins such as phosphoinositide-3 kinase in order to provoke cytokine production and cytotoxicity [89].

### 3.4. HLA-C—A Special Case

Besides the non-classical HLA class Ib genes, HLA-C is the only classical HLA class Ia gene, which is also expressed in extravillous trophoblasts [93]. HLA-C also interacts, according to present knowledge, with different inhibitory receptors, such as KIR2DL1, KIR2DL2, and KIR2DL3, as well as activating receptors, such as KIR2DS2 and KIR2DS1 [69,94,95,96,97]. HLA-C is the only classical HLA class Ia gene, which binds toKIR2DL4, as the non-classical HLA gene HLA-G does. The binding of HLA-C to KIR2DL4 mediates a non-cytotoxic activation of uterine NK cells (uNK, CD 56^bright^), which leads to the synthesis and release of numerous cytokines or growth factors (e.g., G-CSF, GM-CSF, LIF, IL-4, and IL-10), which likewise promote trophoblast and embryo growth [6,7].

### 3.5. Significance of HLA-E to -G for Implantation/Pregnancy

As briefly mentioned above, the placenta or trophoblast cells create an interface between the maternal and embryonal tissue. The relevant cells interacting with the maternal immune system are trophoblast cells, which can be subdivided into cytotrophoblasts (CTs), syncytiotrophoblasts (STs), and EVTs. They emerge from the outer cell mass from early blastocysts and subsequently invade the endometrium. These trophoblast cells are the first to encounter maternal immune cells. During trophoblast invasion, the amount of uNK cells and M2 macrophages, in this context also denoted as decidual macrophages, immediately increases. The uNK cell and macrophage migration might be enhanced by the invading trophoblast cells, which secrete NK cell chemoattractants such as MCP-1 [98]. Uterine NK cells are the dominating immune cells at the beginning of pregnancy, constituting 70–90% of all leukocytes in the decidua [98]. Besides uNK cells, antigen presenting cells (APCs) are the second most abundant population of leukocytes in the decidua [99]. DCs such as DC-10 represent the majority within this population and significantly support the induction of the fetal-maternal immune tolerance. DC-10 cells represent a unique subpopulation, as they express high levels of LILRB1 and -B2 as well as HLA-G. Moreover, they secrete interleukin-10 (IL-10), an immunosuppressive cytokine that induces HLA-G expression [83]. Besides DCs, decidual macrophages also support spiral artery remodeling and placental growth by the secretion of angiogenic growth factors (such as angiogenin, keratinocyte growth factor, fibroblast growth factor B (FGF-B), vascular endothelial growth factor A (VEGF-A), and angiopoietin-1 and -2), matrix metalloproteases (MMP1, -2, -7, -9, and -10), and cytokines (interleukin (IL)-1β, -2, -4, -5, -6, -8, -10, and -13 and TNF-α) [100]. The invading trophoblast interacts with these innate immune cells and T-lymphocytes through the expression of the non-classical HLA groups, here denoted as “embryonic” HLA groups (-E to -G). The expression of the embryonic HLA groups results not only in immune cell inhibition but also in the secretion of the already mentioned growth factors, cytokines and MMPs [101,102]. The HLA expression pattern differs among CTs, STs, and EVTs [15,103]. Moreover, the HLA expression pattern also changes during gestation from implantation to the third trimester [30]. HLA-G and -E are the main HLA class Ib groups expressed during implantation and the first and second trimester to interact with cells of the innate immune system [30]. This interaction supports placentation and the induction of the immune-suppressive milieu to maintain pregnancy. The changes in HLA-G expression across gestation can be recapitulated in plasma samples of pregnant women. During the first trimenon, plasma levels of HLA-G1/G5 increase significantly and thereafter decreases continuously till childbirth [104].

However, with regard to HLA-F, the data concerning expression changes during pregnancy are inconsistent. Shoubu et al. observed that HLA-F is expressed on the surface of EVTs and increases during pregnancy [11], whereas Hackmon et al. demonstrated that HLA-F expression in EVTs decreases [30]. However, Hackmon et al. could also detect an increase of HLA-F expression in the fetal villous mesenchyme during the course of gestation [30].

HLA-E, -F, and -G have also been identified in preimplantation embryos (PIEs) and embryonic stem (ES) cells (also derived from Wharton-jelly) [105,106,107,108]. Preimplantation embryos (Day 3–5) are able to secret extracellular vesicles (EVs) with a diameter of 50–200 nm into the culture media, a phenomenon that has been recently described and named as secretome [109]. The concentration of HLA-G in these EVs is positively correlated with the implantation and pregnancy rate [109,110] (Table 1). In the case of the preimplantation embryo, mainly soluble forms of HLA-G are synthesized [16]. High levels of soluble HLA-G are believed to increase the probability of implantation and higher pregnancy rates in an IVF/ICSI program [18,111,112]. However, these data are still preliminary and a multicenter study demonstrated widely differing results [113]. Moreover, a recently published meta-analysis demonstrates the opposite: the presence of soluble HLA-G in the culture medium results in higher implantation and pregnancy rates [114]. Unfortunately, there is no information which soluble HLA-G isoforms were measured—G5, G6, or G7. However, the knowledge of individual isoform patterns (and polymorphisms) seems to be crucial to better understand the immunological interactions [115]. While HLA-G1 predominates in women with undisturbed pregnancies, women with recurrent spontaneous abortions (RSAs) show higher levels of the short membrane-associated HLA-G4 [116]. RSA patients also show lower levels of soluble HLA-G in maternal serum [117]. This seems to be particularly true for the soluble isoform G5, but not G7 [118]. Moreover, low HLA-G5 levels are often accompanied by low HLA-G1 levels [119].

One possible reason for an aberrant HLA-G protein expression might stem from variations in the genomic sequence of HLA-G. It has been observed that certain SNPs and a 14-pb insertion/deletion in the 3′UTR region influence the HLA-G protein expression. Several studies have found that variations in the 3′UTR, in particular the 3′UTR 14bp deletion [116] or insertion/deletion [115,120,121], increase the risk not only of miscarriage [122,123] but also of repetitive implantation failures (RIFs) [124]. In addition, the *HLA-G* alleles *HLA-G*01:04* and *HLA-G*01:05A* are likewise accompanied by an increased risk of miscarriages [116,125]. Similar results were noted by Vargas et al. for *HLA-G*01:04:01* and *HLA-G*01:01:18* [9] and Hashemi et al. for the G + 3142 > C polymorphism [121]. *HLA*-G gene polymorphisms such as *01:06*, *01:01:06*, *01:01:01:06*, and *01:05N* (null) alleles were significantly higher in patients with RIFs in an IVF/ICSI program [126]. In addition to *HLA-G* polymorphisms, an association between *HLA-E* polymorphisms and RSAs has been observed. The *HLA-E 0101* allele, compared with the *HLA-E 0103* allele, was significantly higher in women suffering from RSAs [127].

Regarding HLA-F, less is known about aberrant expression levels or polymorphisms in RIFs or RSAs.

## 4. “Embryonic“ HLA Genes in Tumors

### 4.1. HLA-G Expression in Cancer

Besides trophoblasts, embryonic HLA genes are also expressed by malignant tumors to escape immune cell recognition and subsequent elimination. This assumption is based on studies investigating the most prominent non-classical HLA gene *HLA-G* [15,36,47,128,129]. Importantly, HLA-G mRNA and protein expression can be found in a broad spectrum of cancer types including breast cancer [130,131,132,133,134,135,136,137], ovarian cancer [79,138,139,140], endometrial cancer [141], cervical cancer [136,142,143,144,145], prostate cancer [146], lung cancer [147,148,149,150,151,152], bladder cancer [153,154,155,156], colorectal cancer [29,35,157,158,159], colon cancer [160], thyroid cancer [161,162], esophageal cancer [163,164], melanoma [165,166,167,168], lymphoma [169], Hodgkin´s lymphoma [170], renal cancer [171,172,173], glioblastoma [174,175,176,177,178], germ cell tumor (testicular) [179] and pancreatic cancer [180] (Table 2). The vast majority of publications show consistently that high expression of class Ib groups, in particular HLA-G, is associated with particularly poor prognosis [163,164,181,182,183,184,185,186]. This may be due to the immune-suppressive property of HLA-G to inhibit proliferation and cytotoxic activity of tumor infiltrated T- and NK cells. Wan et al. reported that HLA-G expression in tumors not only correlates with lower numbers of tumor infiltrating NK cells but also inhibits the cytotoxic potential of residing NK cells probably by binding to their inhibitory receptor LILRB1 [187]. This observation was also found in non-small cell lung cancer, where increased HLA-G and LILRB1 protein expression correlated with increased tumor stage [149]. However, HLA-G interacts not only with LILRB1 but also with the NK cell specific receptor KIR2DL4. Similar to the physiological function during pregnancy, an activation of the KIR2DL4 receptor via HLA-G in tumors not only results in NK cell inhibition but also stimulates the production of growth factors, cytokines, and the release of MMPs. Coexpression of HLA-G and KIR2DL4 therefore worsens cancer prognosis by mediating cancer invasion and metastatic spread, as has been reported in breast cancer [188]. Besides direct immune cell inhibition, an additional indirect mechanism has been shown for the isoforms HLA-G1 and -G5, which are able to induce the generation of HLA-G positive regulatory T-cells (Tregs) and HLA-G+ APCs, such as DC-10 as well as HLA-G positive macrophages (via trogocytosis) [189]. HLA-G+ APCs are able to inhibit the proliferation of CD4+ T-cells and induce the differentiation of CD+ T-cells to Tregs [189,190]. In addition to direct cell–dell interactions HLA-G positive, tumor-associated macrophages (TAMs), can also secrete HLA-G, thereby promoting and supporting an immune-suppressive milieu surrounding cancer cells to further enhance immune evasion. Such TAMs can built up to 50% of the total tumor mass and are consistently associated with a poor prognosis [191]. In addition, tolerogenic DC-10 supports the immune inhibiting effect of TAMs by expressing membrane-bound HLA-G and secreting IL-10. IL-10 can also be synthesized by tumor cells and, in turn, upregulates the expression of HLA-G [192]. The impact of combined expression of IL-10 and HLA-G has been evaluated in lip squamous cell carcinoma (LSCC), where high levels of HLA-G and IL-10 protein expression could be observed only in carcinoma lesions but not in normal tissues [193]. As an additional mechanism, IL-10 induces Type 1 T regulatory (Tr1) cells mediated by the IL-10-dependent ILT4/HLA-G pathway [83].

By mediating an immune-suppressive surrounding, HLA-G counteracts the anti-tumor effect of tumor-infiltrating lymphocytes (TILs). This assumption was supported by Dong et al., who demonstrated that high HLA-G expression was inversely associated with TIL infiltration in breast cancer. Strikingly, they also found that, breast cancer patients with a high HLA-G protein expression and a low TIL infiltration had a significantly higher risk of recurrence compared to patients with low HLA-G expression and a high TIL density [194]. Moreover, it could be shown that inflammatory TILs itself may express HLA-G, as has been demonstrated, e.g., in melanoma [195].

These immune-suppressive effects of HLA-G support the observation that HLA-G expression is associated with poor prognosis. In contrast, some reports have also described opposite associations of HLA-G expression with prognosis in some cancer indications. As one example, Rutten et al. reported that HLA-G protein expression was positively associated with prolonged progression-free survival and improved response to chemotherapeutical treatment in high grade ovarian cancer [139].

However, in all these studies HLA-G protein expression has been has been determined by antibodies detecting all HLA-G isoforms (i.e., clone 4H84, clone MEM-G1 and MEM-G/9). Only a few studies have investigated HLA-G protein isoform expression, by additionally using the anti-HLA-G antibody clone 5A6G7, which recognizes particularly the isoforms HLA-G5/-G6. Zhang et al. evaluated HLA-G5/-G6 expression in ovarian cancer via immunohistochemistry by applying the specific anti-HLA-G antibody clone 5A6G7. Interestingly, they did not find any association of HLA-G5/G6 expression associated with clinical outcome or other variables such as age, histological type, or FIGO stage [138]. In contrast, in non-small-cell lung cancer, immunohistochemical HLA-G5/-G6 expression did discriminate between adenocarcinomas and squamous cell carcinoma [150,196]. It has to be mentioned that that HLA-G isoform expression is not homogeneous within carcinoma entities and mRNA as well as protein expression or synthesis may vary within one tumor [171,196,197]. Besides the known HLA-G isoforms, shed isoforms may occur through proteolytic cleavage by metalloproteinases generating soluble HLA-G1 mediated [198]. Lin et al. assumed the existence of an yet unidentified HLA-G isoform, lacking the alpha 1 domain but containing intron 4 in colon carcinoma [197]. Besides HLA-G isoform expression, polymorphic variants within the DNA sequence such as the 14 bp insertion/deletion in the 3′UTR might also have an impact on the overall increase of cancer risk due to its influence on mRNA stability and protein expression. Li et al. investigated a meta-analysis to evaluate this assumption. The 14 bp insertion/deletion did not significantly correlate with cancer susceptibility, except for breast cancer [199]. A similar observation was published by Haghi et al., demonstrating that the 14 bp deletion was significantly more frequent in patients suffering stage II and III breast cancer than in patients with stage I breast cancer [200].

### 4.2. HLA-E Expression in Cancer

In addition to HLA-G, HLA-E expression can be observed in multiple cancer entities such as breast cancer [132], colorectal cancer [158,159,160,201], renal cancer [173,202,203], lung cancer [204], melanoma [205,206], and gastric cancer [207,208], cervical cancer (adenocarcinoma) [209], glioblastoma [174,177,210], Hepatic carcinoma (hepatocellular) [211], Hodgkin’s lymphoma [212], Thyroid cancer [213] and leukemia [214]. Table 3 shows an overview of tumors in which expression of HLA-E has been detected. Importantly, the occurrence of HLA-E/β2 microglobulin complex, which interacts with the inhibitory NKG2A receptor, has been associated with bad prognosis in colon cancer potentially by contributing to immune evasion of the tumor [159,182]. In addition, HLA-E monomers not bound to β2 microglobulin was also associated with bad prognosis [208]. In line with this, the risk of metastasis is increased in rectal cancer expressing HLA-E [158]. Moreover, the favorable effect on prognosis exerted by TILs is counteracted by a high expression of HLA-E [204,215,216,217]. However, contradictory effects have been observed. Benevolo et al. reported that high HLA-E expression is associated with good prognosis and could therefore serve, in combination with a high expression of HLA-A, as a prognostic marker. They also investigated the expression of the inhibitory NK receptor NKG2A on TILs and determined a positive correlation with HLA-E expression [201]. In renal cell carcinoma HLA-E protein expression determined by IHC was not associated with disease-specific survival, but inversely correlated with the presence of CD56^+^ NK cells [202]. These findings consolidate the immunogenic role of HLA-E based on its interaction with the NK cell receptors CD94/NKG2A, -B, -C, and -D, which may be also expressed in CD8+ T-lymphocytes to lesser extent [8,28,50]. However, based on these data the prognostic value of HLA-E remains to be uncertain. One possible explanation might be the missing analysis of the two functional *HLA-E*0101* and *HLA-E*0103* alleles, which are known to differ in their cell surface expression, thermal stability, and peptide binding affinity [22,23]. Interestingly, Wagner et al. evaluated *HLA-E*0101* and *HLA-E*0103* alleles in patients suffering chronic lymphocytic leukemia. They observed that patients with the *HLA-E*0103* allele are in need of early treatment [218]. However, in renal cell cancer Seliger et al. found no effect on overall survival upon overexpression of HLA-E, while an inverse correlation with tumor infiltrating CD56^+^ NK cells could be demonstrated [202].

### 4.3. HLA-F Expression in Cancer

While several studies have found significant HLA-F expression in several cancer types, the data are even less conclusive. Table 4 lists studies showing HLA-F expression in tumors. HLA-F expression could be observed in breast cancer [219], gastric cancer [207,220], bladder cancer [221], nasopharyngeal cancer [222], esophageal squamous cell cancer [185], hepatocellular carcinoma [186], neuroblastoma [223] and lung cancer [186]. However, no clear correlation with a higher tumor burden or decreased disease specific survival could be drawn. As one example, Zhang et al. observed that HLA-F expression did not correlate with prognosis for gastric cancer patients [224]. However, patients with a positive coexpression of HLA-E and -F suffering gastric cancer had a significantly lower five-year survival rate and a lower postsurgical outcome [207]. In addition, in stage II breast cancer, HLA-F expression correlated with poorer outcome, compared to the HLA-F negative group [219]. The same observation could be found for non-small-cell lung cancer [186] and hepatocellular cancer [184]. One possible reason for the inconsistence of the current data might be that no study distinguishes between the individual isoforms, complex variants, open versus closed conformation and/or antisense polymorphisms. As previously described, HLA-F interacts with LILRB1 and -2 receptors of tumor infiltrating monocytes such as monocytes, DCs, and B-, T-, and NK cells when they migrate into the tumor tissue. However, few data exist regarding the immune-suppressive effect of HLA-F expressed on cancer cells with respect to TILs. Furthermore, contradictory results exist for melanoma patients, where HLA-F expression was not inversely related with the amount of TILs [225].

### 4.4. HLA-C Expression in Cancer

As already described, HLA-C is a special case: as part of the classic HLA complex, it is expressed on nearly every cell type. However, HLA-C is overexpressed on certain tumor cell lines, particularly Cw, which is accompanied with poor prognosis [226,227]. HLA-C primarily activates NK cells (see above), thereby possibly inducing production of cytokines and growth factors that drive malignant cell growth, resulting in tumor promotion. This phenomenon that TILs can also support tumor proliferation rather than limit tumor growth is also well known and contributes to worsen prognosis [191,228,229,230].

### 4.5. Soluble “Embryonic” HLAs in Cancer

The existence of secreted and/or shed embryonic HLA proteins resulting insoluble fractions detected in sera from tumor become of increasing interest. Serum HLA-G levels have been determined in high-grade ovarian, colorectal, gastric, esophageal, lung, and breast cancer, melanoma, and neuroblastoma [140,163,231,232]. Soluble HLA-G (sHLA-G) levels were significantly increased in plasma in breast cancer patients compared to healthy controls and even correlated with the histological type [233]. Interestingly, serum HLA-G levels are associated with estrogen receptor expression and disease progression at the point of diagnosis. This is of particular importance, as estrogen receptor positive “luminal” cancers are well known to have less TILs despite having higher frequencies of multiple signal transduction oncogenes such as PIK3CA compared to estrogen receptor negative tumors. The estrogen dependent HLA gene expression might serve as one underlying mechanism for this phenomenon. Moreover, the high serum level of HLA-G in extracellular vesicles (EVs) correlated with disease progression before neoadjuvant chemotherapy [232]. In lung cancer, sHLA-G in plasma discriminated non-small-cell lung cancer (NSCLC) from small-cell lung cancer (SCLC) and served as a prognostic marker with high levels of sHLA-G indicating reduced overall survival (OS). Similarly, in melanoma, sHLA-G levels were also increased in serum from melanoma patients compared to healthy controls. Furthermore, melanoma patients receiving interferon-alpha (IFN-α) treatment exhibited further increase of sHLA-G levels. Interestingly IFN-α also upregulated HLA-G cell surface expression in circulating monocytes [231]. Morandi et al. corroborated this observation through the detection of sHLA-G secreting monocytes, which had been activated by neuroblastoma tumor cells to secrete sHLA-G [234]. Morandi also evaluated soluble HLA-E (sHLA-E) and -F (sHLA-F) levels in plasma samples from patients suffering from neuroblastoma. Importantly, patients with metastatic disease had higher sHLA-E plasma levels than patients with a localized tumor. In addition to this finding, high sHLA-E and -F levels were observed in relapse-free patients and patients having better OS. [223]. Similar to the findings concerning sHLA-G, sHLA-E levels are also significantly increased in melanoma patients compared to healthy controls [206]. In addition, sHLA-E levels were also significantly elevated in patients suffering from chronic lymphocytic leukemia and were associated with advanced disease stage [218].

In summary, there is consistent evidence in the literature that these HLA groups represent an important factor in determining prognosis [129,142,143,158,160,162,163,216]. This primarily applies to HLA-G and its isoforms, while HLA-E and -F are far less investigated.

### 4.6. Metastases

In clinical tumor therapy, surgical removal of metastases is still an uncommon approach except in rare cases of solitary metastases or oligometastatic disease. Because of that, there are few publications on this issue with regard to HLA expression in metastatic lesions. The expression of HLA-G has been shown in cancer stem cells (CSCs) of certain types of leukemia [235] and renal cancer [236] and in lymph node metastases of thyroid cancer [162], gastric cancer metastases [237], metastases of ovarian cancer [141], malignant melanoma [165,166,205], colon cancer [157], and breast cancer [219,238]. A strong expression of HL-G in the primary tumor increases the likelihood of metastases [128]. However, little is known about the individual isoforms expressed in metastatic tumor tissue. There is evidence of the expression of G1 or the “shed” variant, primarily in colon cancer metastases [128]. Furthermore, HLA-E and -F expression has been found in metastases from breast cancer [132,219].

## 5. Significance for Immunoncology

### General Considerations

Since trophoblast cell invasion and placentogenesis show biological features similar to those of carcinogenesis, malignant tumor invasion and growth could be mediated by similar cellular pathways [3]. It is likely that malignant cells, expressing “embryonic” HLA groups (i.e., HLA class Ib), mediate the same immunological effect as observed in implantation and pregnancy [196]. It has been proven that the expression of “embryonic” HLA groups by tumor cells leads to a similar inhibition of the innate and adaptive immune system and thereby enables tumor immune escape (therapy). This “escape phenomenon” of tumors has long been known and can also be observed in the course of neoadjuvant and adjuvant chemotherapy [22,37,160,168,170,172,178,181,183,239,240,241]. Since the immune evasion mechanism mediated by the expression of immune checkpoints has long been proposed as the major reason for chemotherapy resistance, the investigation of an immune checkpoint therapy has been established. Prominent immune checkpoints are programmed cell death 1 (PD-1) and its ligand (PD-L1), cytotoxic T-lymphocyte associated protein 4 (CTLA4), and indoleamine-2,3-dioxygenase (IDO-1). These immune checkpoints mainly target T-cell response. Anti-PD1 (prembolizumab and nivolumab), anti-PD-L1 (atezolizumab, durvalumab, and avelumab), and anti-CTLA4 (ipilimumab) therapies have been introduced to treat multiple kinds of cancer. However, while having provided substantial promise as a new treatment strategy, the modulation by immune checkpoint therapies are only successful in 15–20% of patients [241,242,243,244,245,246]. One reason for this limited activity of checkpoint inhibitors might be the simultaneous expression of embryonic HLA genes on cancer cells, which govern an independent immune inhibition mechanism. Early hints in this direction stem from studies of intraoral mucoepidermoid carcinomas, where HLA-G, HLA-E, and PD-L1 expression was determined via immunohistochemistry [247]. Similarly, Lopes et al. observed coexpression of PD-L1 and HLA-G proteins in lip carcinomas. This coexpression was associated with a higher malignancy and occurrence of systemic metastases [246]. Therefore, it is conceivable that, when confronted with protective mechanisms similar to the defense mechanisms in pregnancy, immune oncological therapeutic approaches are less effective than expected [248]. It has to be noted that, compared to trophoblastic cells, cancer cells express the classical HLA class Ia genes (*HLA-A, -B*, and *-C*) and therefore escape NK cell recognition and elimination through the interaction with KIRs [249]. These conditions are fundamentally different from pregnancy, where the embryo has foreign antigenicity and needs more intensive protection.

## 6. Hypotheses for an Immunological Tumor Therapy Concept (ITTC)

The patent focuses on determining the individual expression pattern of the new immune-oncological targets to dissect the cellular communication of specific malignant cell clones with the innate and adaptive immune system (Figure 4). This shall lead to tailored immune oncological treatment modalities to break the resistance of tumors refractory to check point inhibition. These assumptions are based on our detailed knowledge from implantation and pregnancy, where the communication has to be regarded as multifactorial and depends on the physiologic expression of non-classical HLA class Ib isoforms to prevent spontaneous abortions and enable implantation, growth and development of the embryo. Apparently, the communication based on non-classical HLA genes is extremely successful and is crucial to allow the genetically (semi-) foreign embryo or fetus to grow undisturbed despite close cellular interdigitation.

In their untreated state, malignant tumors are also “successful” in terms of unrestricted growth, immune escape, and “implantation” into distant organs. They certainly use multifactorial—not monofactorial—cell communication by expression of various types of immune inhibiting factors including the non-classical HLA class Ib gene, while not being restricted to check point modulators. The resulting immune inhibition and evasion is based on a landscape of multiple inhibiting receptors expressed in cells of the innate and adaptive immune system by far exceeding the check point receptors. Therefore, the identification and determination of the embryonic HLA groups expression pattern on malignant tumors (including isoforms) defines the extent of the underlying immune cell inhibition and communication. However, this in turn means that any monofactorial anti-tumor treatment may be ineffective in most situations. For this reason, an extensive HLA profiling including robust target quantitation for every malignant tumor seems to be necessary. Importantly, it is reasonable to monitor the HLA expression patterns for subsequent treatment adaptations as metastases are known to be able to modify their biology. As has been observed in breast cancer, metastasis can frequently convert from being an estrogen-receptor-positive primary tumor into an estrogen-receptor-negative metastasis [250,251]. As hormone receptors i.a. also control HLA gene expression this has to be taken into account. In line with this, HLA expression or the underlying gene activation also frequently changes during disease progression [132,165,252]. Because of that, HLA profiling including robust quantitation of individual HLAs should be repeated in cases of metastases (“metastatic surgery”).

According to the initial companion diagnostic concepts in breast cancer, where Her-2/neu status has to be determined before tailored therapy regimen are given, an adopted approach for the embryonic HLA genes seems to be reasonable. If a tumor/metastasis expresses “embryonic” HLA groups or specific isoforms treatment by tailored antibodies is a promising option. Human antibodies against HLA-E, -F, and -G are partly already commercially available [42].

Specific antibody treatment should be administered based on the results of HLA profiling. The following treatment strategies/concepts are possible: Masking antibodies mask the corresponding HLA groups/isoforms, blocking the “escape mechanism” of the tumor cells and allowing the immune system to attack the tumor as normal.The concept of special antibodies, e.g., coupled with receptors, weak radiation, or chemotherapy drugs, is based on activating the defenses of the immune system as well as attacking the tumor cells directly.As radiation and, to a lesser extent, chemotherapy drugs can change the steric structure of an antibody, the antibody would be administered first, followed by an application of the radiation particle or the drug.

The underlying rationale of this patent offers several further treatment possibilities. Instead of applying antibodies, in situ gene editing could be performed to erase the expression of non-classic HLA groups (patent pending). Another approach is the blockage of the corresponding receptors on NK cells and lymphocytes, similar to the concept of, e.g., the PD-1/PD-1-ligand therapeutic system (patents pending). Furthermore, vaccination/immunization approaches by injecting non-viable tumor cells or their membranes (if expressing neo-antigens) could be a further therapeutic concept. The strategy of this approach is the blocking or eradication of non-classical HLA groups on the surface of the non-viable tumor cells prior to injection (patent pending) to enable recognition and antibody generation against the exposed neo-antigens. Based on the assumption that the HLA system may dominate other immune escape mechanisms, it will be intriguing to track clinical approaches evaluating individual HLA proteins as new anti-cancer targets with striking potential synergy to current check point inhibitors. However, for this purpose, a robust quantitation of the individual HLA profile seems to be crucial to reduce potential side effects and tailor the most important treatment option.

## Figures and Tables

**Figure 1 ijms-20-01830-f001:**
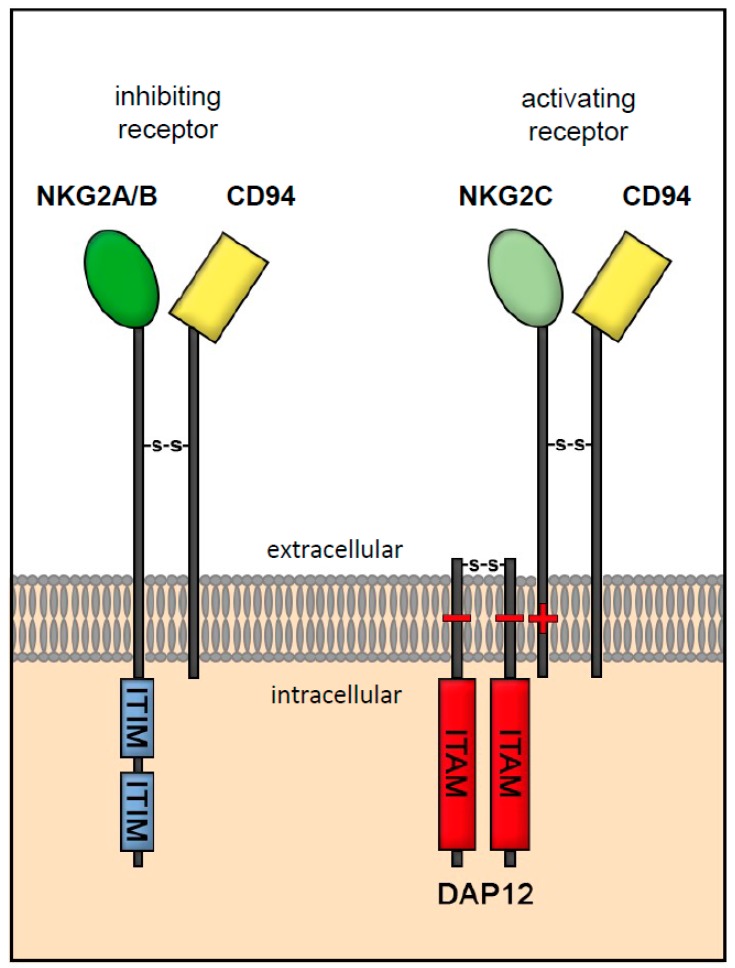
Receptor interaction of HLA-E with NKG2A, -B/CD94 and NKG2C/CD94. HLA-E binds to the inhibiting receptors NKG2A, -B and activating receptor NKG2C, belonging to the killer cell lectin-like receptor C1 (KLRC1) family, expressed on NK cells. The NKG2A and -B receptors mediate an inhibitory signal to the NK cell via immunoreceptor tyrosine-based inhibition motifs (ITIMs) [51]. The activating receptor NKG2C does not possess an intracellular immunoreceptor tyrosine-based activating motif (ITAM), but contains a positively charged transmembrane domain and dimerize with DNAX activation protein 12 (DAP-12), which has an ITAM in its cytoplasmic domain and transmits an activating signal to the cell [51,54,55].

**Figure 2 ijms-20-01830-f002:**
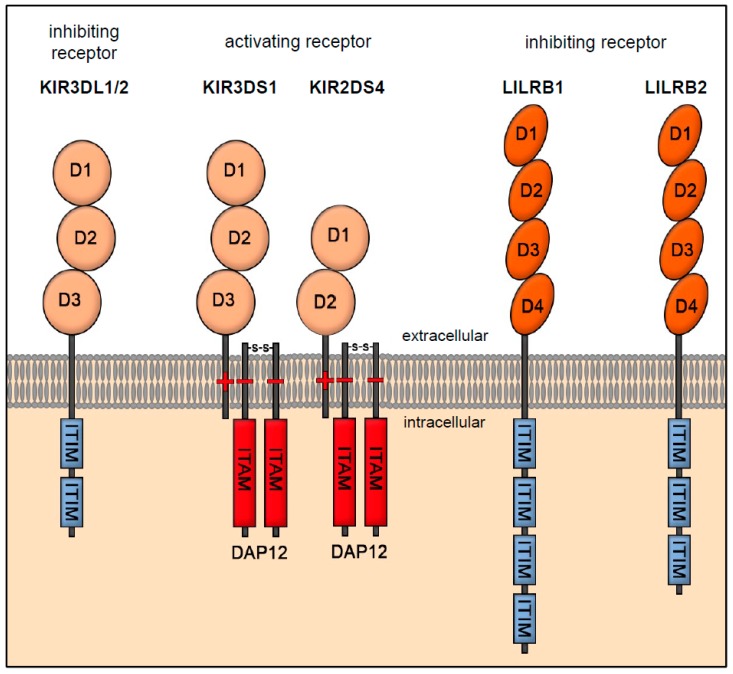
Receptor interaction of HLA-F with KIR3DL1, -2, KIR3DS1 and -S4 as well as LILRB1 and -B2. The inhibiting receptors KIR3DL1 and -2 and activating receptors KIR3DS1 and KIR2DS4 belong to the family of killer cell immunoglobulin-like receptors (KIRs), which are expressed on NK cells. The inhibiting receptors KIR3DL1 and -2 have a long cytoplasmatic tail (L) with immunoreceptor tyrosine-based inhibition motifs (ITIMs). The activating receptors KIR3DS1 and KIR2DS4 are classified by their number of extracellular domains (two or three domains, 2D or 3D) and short (S) intracellular cytoplasmatic tail, which contains a charged lysine residue instead of an immunoreceptor tyrosine-based inhibition motif (ITIM). They dimerize with DNAX activation protein 12 (DAP-12), which has an immunoreceptor tyrosine-based activating motif (ITAM). The inhibitory leukocyte-immunoglobulin (Ig)-like receptors (LILR) LILRB1 (also known as Ig-like transcript 2; ILT2) and -B2 (also known as ILT4) are expressed on monocytes, dendritic cells (DCs), as well as on B-, T-, and NK cells and mediate an inhibitory signal via their ITIMs.

**Figure 3 ijms-20-01830-f003:**
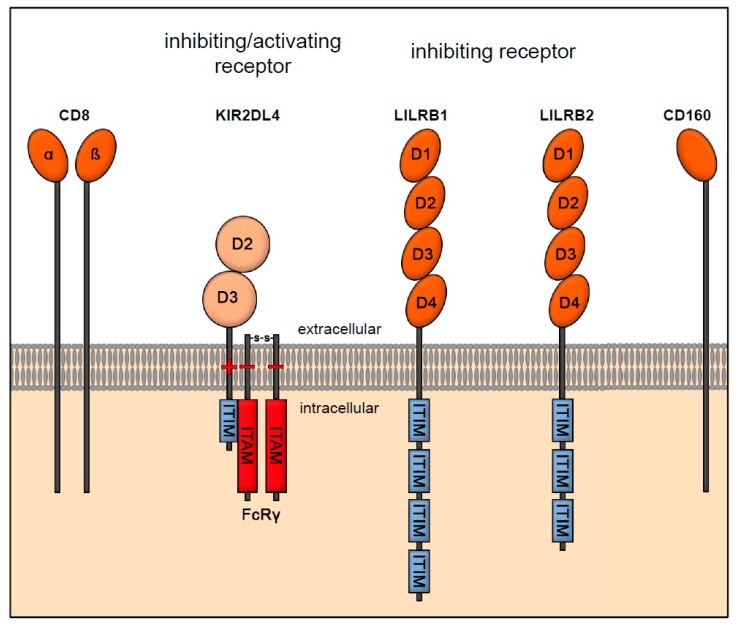
Receptor interaction of HLA-G with CD8, KIR2DL4, LILRB1, LILRB2 and CD160. CD8 is a marker for cytotoxic T-cells and consists of an extracellular alpha and beta domain [88]. The receptor KIR2DL4 belongs to the family of killer cell immunoglobulin-like receptors (KIRs), which are expressed on NK cells. The receptor contains two extracellular domains D0 and D1 and has only one immunoreceptor tyrosine-based inhibition motifs (ITIM). A charged arginine residue in its cytoplasmatic tail enables KIR2DL4 to form a complex with Fc fragment receptor γ (FcRγ), which stimulates cytokine production in the NK cell [55]. The inhibitory leukocyte-immunoglobulin (Ig)-like receptors (LILR) LILRB1 (also known as Ig-like transcript 2; ILT2) and -B2 (also known as ILT4) are expressed on monocytes, dendritic cells (DCs), as well as on B-, T-, and NK cells and mediate an inhibitory signal via their ITIMs [36,39,74]. CD160 is a glycosylphosphatidylinositol-anchored receptor and does not contain an immunoreceptor tyrosine-based activating motif (ITAM) [89].

**Figure 4 ijms-20-01830-f004:**
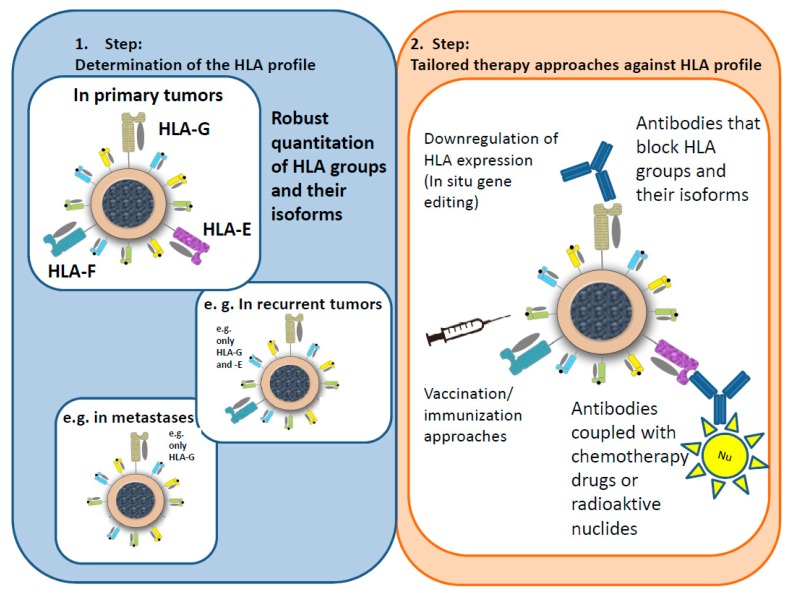
Hypotheses for an immunological tumor therapy concept (ITTC). The patent focuses on determining the individual expression pattern of the “embryonic” HLA genes on primary tumors to create individual therapy approaches such as antibodies drug conjugates, vaccination/immunization approaches and in situ gene editing to block and downregulate HLA class Ib expression in order to overcome immune evasion mediated by the non-classical HLA groups. The patent also implies the monitoring of the HLA expression patterns on recurrent tumors and metastases for subsequent treatment adaptations.

**Table 1 ijms-20-01830-t001:** Overview of proven occurrences of HLA groups E-G and their locations.

Localization	HLA Gene	Authors
CTs, STs, EVTs	HLA-E, HLA-F, HLA-G	Apps et al., 2008 [15] Hackmon et al., 2017 [30] Rizzo et al., 2011 [103] Shobu et al., 2006 [11]
ES	HLA-E, HLA-G	Drukker et al., 2002 [105]
PIE	HLA-G	Fuzzi et al., 2002 [106] Shaikly et al., 2008 [16]
	sHLA-G	Sher, 2004 [112] Tabiasco et al., 2009 [113]
PIE/ES	HLA-G	Rizzo et al., 2011 [103] Verloes et al., 2011 [107]
ES (Wharton-jelly)	HLA-E, HLA-F, HLA-G	Chen et al., 2012 [108]
Culture medium of PIE	sHLA-G	Noci et al., 2005 [110]

CTs = cytotrophoblasts, ES = embryonic stem cells, EVTs = extravillous trophoblasts, PIE = preimplantation embryo, STs = syncytiotrophoblasts.

**Table 2 ijms-20-01830-t002:** Selected literature on expression of HLA-G in malignant tumors.

Carcinoma	Authors
Bladder cancer	Castelli et al., 2008 [154] El-Chennawi et al., 2005 [155] Gan et al., 2010 [156]
Breast cancer	Jeong et al., 2014 [130] Rolfsen et al., 2013 [131] Da Silva et al.,2013 [132] He et al.,2010 [133] Kleinberg et al., 2006 [134] Palmisano et al., 2002 [135] Singer et al., 2003 [136] de Kruijf et al., 2010 [137]
Cervical cancer	Gimenes et al., 2014 [142] Li et al., 2012 [143] Rodriguez et al., 2012 [144] Zheng et al., 2011 [145] Ferguson et al., 2012 [136]
Colon cancer	Zeestraten et al., 2014 [160]
Endometrial cancer	Bijen et al., 2010 [141]
Esophageal cancer	Cao et al., 2011 [163] Lin et al., 2011b [164]
Germ cell tumor (testicular)	Karagoz et al., 2014 [179]
Glioblastoma	Kren et al., 2010 [174] Kren et al.,2011 [175] Wastowski et al., 2013 [176] Wiendl et al., 2002 [177] Wischhusen et al., 2007 [178]
Hodgkin’s lymphoma	Diepstra et al., 2008 [170]
Lymphoma	Urosevic et al., 2002 [169]
Lung cancer	Montilla et al., 2016 [147] Urosevic et al., 2001 [148] Wisniewski et al., 2015 [149] Yan et al., 2015 [150] Yie et al., 2007 [151] Zhang et al., 2016 [152]
Malignant melanoma	Degenhardt et al., 2010 [165] Bezuhly et al., 2008 [166] Paul et al., 1999 [167] Paul et ak., 1998 [168]
Ovarian cancer	Zhang et al., 2016 [138] Rutten et al., 2014 [139] Lin et al., 2013 [140]
Pancreatic cancer	Zhou et al., 2015 [180]
Prostate cancer	Langat et al., 2006 [146]
Rectal cancer	Reimers et al., 2014 [163] Guo et al., 2015 [162]
Renal cancer	Tronik-Le Roux, 2017 [171] Ibrahim et al., 2001 [172] Hanak et al., 2009 [173]
Thyroid cancer	Dardano et al., 2011 [161] Nunes et al., 2013 [162]

**Table 3 ijms-20-01830-t003:** Selected literature proving expression of HLA-E on malignant tumors.

Carcinoma	Authors
Breast cancer	da Silva et al, 2013 [132]
Cervical carcinoma (adenocarcinoma)	Spaans et al., 2012 [213]
Colorectal cancer	Guo et al., 2015 [158]
	Reimer et al., 2014 [159]
	Zeestraten et al., 2014 [160]
	Benevolo et al., 2011 [201]
Gastric cancer	Ishigami et al., 2015 [207]
	Sasaki et al., 2014 [208]
Glioblastoma	Kren et al., 2011 [174] Wischhusen et al., 2007 [178] Wolpert et al., 2012 [210]
Hepatic carcinoma (hepatocellular)	Chen et al., 2011 [211]
Hodgkin’s lymphoma	Kren et al., 2012 [212]
Leukemia	Xu et al., 2018 [214]
Lung cancer	Talebian et al., 2015 [204]
Melanoma	Tremante et al., 2014 [205]
	Allard et al., 2011 [206]
Renal cancer	Hanak et al., 2009 [173] Seliger et al., 2016 [202] Kren et al., 2012 [203]
Thyroid cancer	Zanetti et al., 2013 [213]

**Table 4 ijms-20-01830-t004:** Studies showing expression of HLA-F in malignant tumors.

Carcinoma	Author
Breast cancer	Harada et al., 2015 [219]
Bladder cancer	Li et al., 2018 [221]
Esophageal cancer	Zhang et al., 2013 [185]
Gastric cancer	Ishigami et al., 2015 [207]
	Ishigami et al., 2013 [220]
	Zhang et al., 2013 [224]
Hepatic cancer (hepatocellular)	Xu et al., 2015 [184]
Lung cancer	Lin et al. 2011a [186]
Neuroblastoma	Morandi et al., 2013 [223]

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
