# Peer review of "European Patent in Immunoncology: From Immunological Principles of Implantation to Cancer Treatment"

_ijms, 2019, doi:10.3390/ijms20081830_

Reviewer 1 Report

The review article submitted by Würfel et al. focuses on the background of a filed patent regarding embryo implantation immunological principles and the use of this knowledge for immunological cancer treatment. Most parts of the review highlight the findings from reproductive biology/medicine and corresponding research in various types of cancer. Overall, the review is clearly structured, clearly written and very detailed. The part about the potential applications is a bit short. An outlook could also include additional treatment possibilities such as in situ gene editing to manipulate expression of non-classical HLA genes.

The figures were missing in the PDF.

Title: Is the title derived from the patent? “European patent in immunoncology: from immunological principles of implantation based to cancer treatment” Maybe the word “based” should be deleted.

Line 43: Maybe better write: “…immune system, but the trophoblast respectively placenta.”

Line 55: Why in contrast to viviparity? To my knowledge, humans belong to the organisms showing viviparity, i.e., placental viviparity.

Line 56: “Because of this special situation…” could be specified. The fetal tissues have to escape the maternal immune system to maintain pregnancy. In pregnancy failure and disorders this is not working properly. Furthermore, this mechanism is used by tumor cells. This makes the investigation of non-classical HLA interesting.

Line 63ff: Use code for or encode

Line 71: It should be either “signal peptides” or “leader sequences” but not both. The cited reference is not the original work.

Line 100: “High expression…”

Line 103: Write “most prominent gene” or “bets known gene”

Line 154ff: Since the HFE gene is not HLA-H, this should be deleted.

Author Response

We thank the reviewer for his helpful comments.

“The figures were missing in the PDF.”

The reviewer mentioned the missing figures in the peer review version. Since we had to upload the figures separately, we did not insert the figures in the original paper version.

“Title: Is the title derived from the patent? “European patent in immunoncology: from immunological principles of implantation based to cancer treatment” Maybe the word “based” should be deleted.”

The second comment has been addressed to the titel, concerning the word “based”. We deleted this word, as suggested by the reviewer.

 “Line 43: Maybe better write: “…immune system, but the trophoblast respectively placenta.”

In Line 45 (Line 43 in the reviewers report), the reviewer recommended the rearrangement of the sentence. We also rearranged the sentence in Line 45 from “In pregnancy, trophoblast, respectively the placenta creates an “interface” between the embryo/fetus and the maternal immune system” to “In pregnancy the placenta more specifically, trophoblast, creates an “interface” between the embryo/fetus and the maternal immune system”

“Line 55: Why in contrast to viviparity? To my knowledge, humans belong to the organisms showing viviparity, i.e., placental viviparity.”

We deleted the described sentence “It should be noted that implantation, i.e. pregnancy in humans, is a close cellular “network” between embryonic and maternal tissues; this is in contrast to viviparity, in which the offsprings develop in utero without such a cellular network.”  in Line 60-62 (Line 55 in the reviewers report), which included the reviewers argument about viviparity, since the  term viviparity in the context of the senctence was slightly misleading and the sentence itself is not necessary for the overall understanding of the paper.

“Line 56: “Because of this special situation…” could be specified. The fetal tissues have to escape the maternal immune system to maintain pregnancy. In pregnancy failure and disorders this is not working properly. Furthermore, this mechanism is used by tumor cells. This makes the investigation of non-classical HLA interesting.”

We agree with this comment from the reviewer, since the introduction does not really introduce into the topic of the paper. Therefore, from line 63 to 70, we rearranged this section, as recommended by the reviewer in order to better introduce and specify the topic of the paper. In detail, we mentioned the expression of the non-classical HLA groups on trophoblast and their interaction with the adaptive and innate immune system by interacting with killer immunoglobulin-like receptors (KIR) and lymphocyte immunoglobulin like receptors (LIL-R).  Since tumor cells use the same immune escape mechanism, we wanted to address this knowledge to the reader to highlight the unique immune inhibition properties of the non-classical HLA genes.

“Line 63ff: Use code for or encode”         and

“Line 71: It should be either “signal peptides” or “leader sequences” but not both. The cited reference is not the original work.”

Regarding the section about HLA-E (Line 75 to 93) we did some minor corrections, as suggested by the reviewers concerning spelling; “encode” instead of “code” (line 76,77), deletion of “s” after “for” and of “leader sequence”. We also checked the reference and updated the cited reference.

“Line 100: “High expression…”

The remark from the reviewer concerning this point was slightly unclear to us, we assumed that this comment might refer to spelling.  However, we want to point out that the phrase “high” expression in regards to the general HLA-F expression is quite imprecise and could be deleted, since HLA-F protein expression at the cell surface is mainly restricted to trophoblast and activated B-cells. Therefore, the phrase “high” does not relate to a normal protein expression.

 “Line 103: Write “most prominent gene” or “bets known gene””

In line 119 we deleted the words “and known”

“Line 154ff: Since the HFE gene is not HLA-H, this should be deleted.”

From Line 163 to 177 we decided to delete the whole section about HLA groups H to Z, since this part about the HFE gene and HLA-H could be confusing to the reader. Additionally, this part is not mandatory for understanding the paper and the patent itself.

“….The part about the potential applications is a bit short. An outlook could also include additional treatment possibilities such as in situ gene editing to manipulate expression of non-classical HLA genes.”

As recommended by the reviewer, the part about hypotheses for an immunological tumor therapy concept (ITTC) has been enlarged (line 623-630) by describing further possible treatment options. We hope that the inserted sections in this part meet the reviewers idea concerning the outlook. We kept the section about additional treatment possibilities general, since the patent does not contain detailed technical descriptions about gene editing.

Reviewer 2 Report

The authors have reviewed the literature relevant to a patent related to the immunological treatment of cancer. This treatment is based on the hypothesis that tumors hijack a family of potent inhibitors of the immune system known to induce immune tolerance during pregnancy. The authors argue that the aberrant expression of these non classical HLA molecules in tumors is key in creating an immune privilege environment and determine a worse prognosis. They go on a propose that HLA profiling should thus be performed on every malignant tumor, and then treated with a antibodies against those molecules.

Although the hypothesis is supported by bibliography, and the review is generally well-structured, the authors should address several points that affect the overall roundness of the review and could contribute to a better reading experience.

First is the English use of certain words. For example, the use of the word respectively throughout the review should be substituted by and/or. In English it means in the order already mentioned, and it is frequently misused in the text.  Examples are line 15, 43, 49, 134, 262, etc.  

In line 55, the authors oppose pregnancy in humans to viviparity, while the former could also be defined as placental viviparity.

In line 105 please correct almoust (almost).

In line 106, please comma after At present.

In line 170, 194 and 225, please substitute the non-existing word receptorinteraction with Receptor interaction/s.

In line 173, substitute represent for represents.

In line 176, remove merely.

In line 290, rewrite the sentence containing opposite (removing it) and the two previous one to facilitate the reading, avoiding the current structure (contradicting twice the previous sentences).

In line 295, remove 2013)

In line 302, please insert “that” before variations.

In line 310 substitute Embronic for Embryonic.

In line 312, substitute on for in.

In line 318, table 4 appears and should follow the logical order and thus be table 2.

In line 343 and 344, the sentences saying “ This immune suppressive effects of HLA-G verifies and explains the oberservation that HLA-G expression is associated with poor prognosis. However, only a few puplications reported a vice versa effect of HLA-G expression” could be something like “The(se) immune suppressive effects of HLA-G support the observation that HLA-G expression is associated with poor prognosis. However, some reports have also described other instances where the opposite effect of HLA-G expression was observed”

In line 353, acociated should be associated.

In line 354, substitute “This observation could be disproven for” for “ However, in” and then remove “where” before immunohistochemical.

In line 359, substitute seeded for shredded.

In line 372, substitute Tab. 2 by Table 3 (and change the corresponding table in the supplementary).

In line 375, substitute escape by immune evasion, for example. When written ' In case of unbound HLA-E monomers without ß2 microglobulin worsen prognosis” please substitute by “When unbound HLA-E monomers without ß2 microglobulin are present prognosis worsens”

In line 382, For renal cell carcinoma,  immunhistochemically…

In 387, “ The controversial finding for HLE” for “These findings”.

In 396, “ fever and controversal data exist” for “fewer and less clear data is available”

In 401, remove comma.

In 408 substitute even for appear.

In 434 insert a before marker and eliminate discrimination.

In 438, insert comma after In addition to that.

In 441, change verfy for corroborate.

In 450 remove it can be stated, and change consens for consensus.

In 455, insert a comma after Because of that.

In 467, remove it can be assumed and substitute is by could be.

In 468 substitute It can be expected by It is likely.

In 478 substitute majorly by mainly.

In 486, please eliminate space and full stop.

In 495 eliminate a.

In 502, substitute gens by genes.

In line 496, a figure is mentioned but never shown. Actually, it would be extremely helpful for the reader to have a figure/figures where the basic hypothesis is presented highlighting the most important mechanistic details that are considered relevant by the authors.
Although the focus is clear and appreciated, the authors state that the non-classical HLA genes predominantly mediate immune evasion by interacting with different receptor present on different compartments of the innate and adaptive immune system. They present some examples, like induction of apoptosis of CD8-positive cells, but fail to mention others, like suppression of proliferation in CD4+ T cells (Bainbridge et al. 2000) or induction of Tregs (Selmani et al. 2008). Being a review that aims at giving supporting evidence for the effects of the non-classical HLA molecules, a table with a more comprehensive list of cellular targets for the different HLA molecules and corresponding references would also be appreciated.

Author Response

“First is the English use of certain words. For example, the use of the word respectively throughout the review should be substituted by and/or. In English it means in the order already mentioned, and it is frequently misused in the text.  Examples are line 15, 43, 49, 134, 262, etc. “

 In order to eliminate the misuse of the word respectively, we replaced “respectively” by “more specifically” (line 15), “and/or” (line 47), “specifically” (line 115), “and” (line 152), “or” (line 307). In line 52 we deleted the passage “…embryo respectively the…”.

“In line 55, the authors oppose pregnancy in humans to viviparity, while the former could also be defined as placental viviparity.”

This point has also been addressed by the first reviewer. We deleted the described sentence “It should be noted that implantation, i.e. pregnancy in humans, is a close cellular “network” between embryonic and maternal tissues; this is in contrast to viviparity, in which the offsprings develop in utero without such a cellular network.”  in Line 60-62 (Line 55 in the reviewers report), which included the reviewers argument about viviparity, since the  term viviparity in the context of the senctence was slightly misleading and the sentence itself is not necessary for the overall understanding of the paper.

“In line 105 please correct almoust (almost).”

We replaced “almoust” by “almost” (in the revised paper line 121, line 105 in the reviewers report).

“In line 106, please comma after At present.”

We inserted a comma after “At present” according to the reviewers recommendation (line 122, but line 106 in the reviewers report).

“In line 170, 194 and 225, please substitute the non-existing word receptorinteraction with Receptor interaction/s.”

In line 187 (170 in the reviewers report), 216 (line 194 in the reviewers report) and line 249 (line 225 in the reviewers report) we inserted a space character to change the word “receptorinteraction” to “receptor interaction”.

“In line 173, substitute represent for represents.”

We added an “s” after “represent” to change the word from “represent” to “represent”, as recommended by the reviewer (line 190 in the revised paper, but line 173 in the reviewers report).

“In line 176, remove merely.”

We deleted the word “merely” in line 193 (line 176 in the reviewers report).

“In line 290, rewrite the sentence containing opposite (removing it) and the two previous one to facilitate the reading, avoiding the current structure (contradicting twice the previous sentences).”

We need to apologize regarding this comment from the reviewer, since we could not directly implement the remark. However, we rearranged the sentence in line 300-301to achieve more clearness in this section regarding the content. 

“In line 295, remove 2013)”

In line 364 of the revised paper (line 295 in the reviewers report), we removed “2013)” to meet the reviewers remark.

“In line 302, please insert “that” before variations.”

In line 374 (line 302 in the reviewers report) we inserted “that” before “variations”, according to the reviewers suggestion.

“In line 310 substitute Embronic for Embryonic.”

We inserted an “y” in line 387 (line 310 in the reviewers report) in the word “Embronic” to change it to the word “Embryonic” to meet the reviewers remark.

“In line 312, substitute on for in.”

In line 390 (line 312 in the reviewers report) we replaced “in” by the word “on”, according to the reviewers comment.

“In line 318, table 4 appears and should follow the logical order and thus be table 2.”

We changed the numeration of the table from “4” to “2” to follow the logical order.

In line 343 and 344, the sentences saying “ This immune suppressive effects of HLA-G verifies and explains the oberservation that HLA-G expression is associated with poor prognosis. However, only a few puplications reported a vice versa effect of HLA-G expression” could be something like “The(se) immune suppressive effects of HLA-G support the observation that HLA-G expression is associated with poor prognosis. However, some reports have also described other instances where the opposite effect of HLA-G expression was observed”

In line 442 to 448 (line 343 and 344 in the reviewers report), we deleted the sentences “This immune suppressive effects of HLA-G verifies and explains the observation that HLA-G expression is associated with poor prognosis. However, only a few publications reported a vice versa effect of HLA-G expression.” We rearranged this section, according to the reviewers comment to “These immune-suppressive effects of HLA-G support the observation that HLA-G expression is associated with poor prognosis. However, in contrast, some reports have also described other instances where the opposite effect associations of HLA-G expression was observed with prognosis in some cancer indications. Rutten et al. reported that HLA-G protein expression was positively associated with prolonged, better, and progression-free survival and even with better improved response to chemotherapeutical treatment in high grade ovarian cancer.” in order to achieve a clearer structure and reading experience.

“In line 353, acociated should be associated.”

In line 459 (line 353 in the reviewers report) we replaced “acociated” by “associated”.

“In line 354, substitute “This observation could be disproven for” for “ However, in” and then remove “where” before immunohistochemical.”

According to the reviewers remark, we deleted “This observation could be disproven for” and inserted “In contrast, in”. We also deleted the word “were” before “immunohistochemical” (line 460-461 in the revised paper, but line 354 in the reviewers report).

“In line 359, substitute seeded for shredded.”

In line 467 (line 359 in the reviewers report) we substituted the word “sheeded” by the word “shedded” to meet the reviewers’ comment.

“In line 372, substitute Tab. 2 by Table 3 (and change the corresponding table in the supplementary).”

We changed the numbering of the table from “2” to “3” to follow the logical order. Additionaly, the substituted “Table” for “Tab.”, as recommended by the reviewer.

“In line 375, substitute escape by immune evasion, for example. When written ' In case of unbound HLA-E monomers without ß2 microglobulin worsen prognosis” please substitute by “When unbound HLA-E monomers without ß2 microglobulin are present prognosis worsens””

From line 487 to 490 (line 375 in the reviewers report), we did some changes as suggested by the reviewer; we exchanged “escape” by “immune evasion” and eliminated the sentences “In case of unbound HLA-E monomers without ß2 microglobulin worsens prognosis” and replaced it by “ In addition HLA-E monomers not bound to ß2 microglobulin, was also associated with bad prognosis” to achieve a better syntax. 

“In line 382, For renal cell carcinoma,  immunhistochemically…”

Regarding the sentence in line 497 to 499 (line 382 in the reviewers report) “For renal cell carcinoma, immunhistochemically determined HLA-E protein expression is not associated with disease specific survival, but an inverse correlation with CD56+ NK cells could be observed” has been changed to “In renal cell carcinoma, HLA-E protein expression determined by IHC was not associated with disease specific survival, but inversely correlated with the presence of CD56+ NK cells.” to achieve also a better syntax, as recommended by the reviewer.

In 387, “ The controversial finding for HLE” for “These findings”.

Regarding this remark, we deleted the part “The controversial finding for HLA” in line 504 (line 387 in the reviewers report) and replaced it by “One explanation could be explained by the missing analysis of the two functional HLA-E*0101 and HLA-E*0103 alleles, which are known to differ in their cell surface expression, thermal stability and peptide binding affinity.” to improve the syntax in this section as suggested by the reviewer.

“In 396, “ fever and controversal data exist” for “fewer and less clear data is available””

As recommended by the reviewer, we removed “fever and controversial data exist” in line 516 (line 396 in the reviewers report) and inserted “these data are even less conclusive”.  

“In 401, remove comma.”

In line 522 (line 401 in the reviewers report) we removed the comma after the word “observed”.

“In 408 substitute even for appear.”

In line 531 (line 408 in the reviewers report) we deleted the whole sentence “As these are assumed to have different biological effects, the results of the studies are difficult to compare or are even contradictory”, which contained the word “even”.

“In 434 insert a before marker and eliminate discrimination.”

To meet the reviewers recommendations (line 567 in the revised review, but line 434 in the reviewers report), we did not only eliminate “discrimination” but also rearranged the whole sentence to “In lung cancer, sHLA-G in plasma discriminated non-small-cell lung cancer (NSCLC) from small-cell lung cancer (SCLC) and served as a prognostic marker with high levels of sHLA-G indicating reduced overall survival (OS).”

“In 438, insert comma after In addition to that.”

In line 573 (line 438 in the reviewers report) we inserted a comma after the word “Furthermore”.

“In 441, change verfy for corroborate.”

We substituted “verfy” by the word “corroborate” (line 576 in the revised paper, but line 441 in the reviewers report).

“In 450 remove it can be stated, and change consens for consensus.”

In line 587 in the revised paper (line 450 in the reviewers report), we changed “It can be stated that” to “In summary,” and “consens” to “consistent evidence”.

“In 455, insert a comma after Because of that.”

In line 593 in the revised paper (line 455 in the reviewers report) we inserted a comma after the phrase “Because of that”.

“In 467, remove it can be assumed and substitute is by could be.”

We removed the part “it can be assumed that” in line 607 (line 467 in the reviewers report) and exchanged “is” by “ it could be” as recommended by the reviewer.

“In 468 substitute It can be expected by It is likely.”

We exchanged “can be expected” by “is likely” in line 608 (line 468 in the reviewers report) as mentioned by the reviewer.

“In 478 substitute majorly by mainly.”

In line 619 in the revised paper (line 478 in the reviewers report) we replaced “majorly” by “mainly” as remarked by the reviewer.

“In 486, please eliminate space and full stop.”

In line 631 of the revised paper (line 486 of the reviewers report), we deleted the full stop and the space.

“In 495 eliminate a.”

We deleted the “a” in line 642 of the revised paper (line 495 of the reviewers report).

“In 502, substitute gens by genes.”

In line 655 (line 502 in the reviewers report), we added an “e” to “gen” to change the word to “genes” as recommended by the reviewer.

“In line 496, a figure is mentioned but never shown. Actually, it would be extremely helpful for the reader to have a figure/figures where the basic hypothesis is presented highlighting the most important mechanistic details that are considered relevant by the authors.”

For the original paper version we prepared figures depicting the corresponding receptors of the non-classical HLA genes (figure 1 to 3) and a figure regarding the principle of the patent. The first reviewer also remarked this point, but since we were instructed to upload the figures separately, we did not insert the figures in the original paper version.

“Although the focus is clear and appreciated, the authors state that the non-classical HLA genes predominantly mediate immune evasion by interacting with different receptor present on different compartments of the innate and adaptive immune system. They present some examples, like induction of apoptosis of CD8-positive cells, but fail to mention others, like suppression of proliferation in CD4+ T cells (Bainbridge et al. 2000) or induction of Tregs (Selmani et al. 2008). Being a review that aims at giving supporting evidence for the effects of the non-classical HLA molecules, a table with a more comprehensive list of cellular targets for the different HLA molecules and corresponding references would also be appreciated.”

We do understand the remark from the reviewer, since the immunological aspect has been slightly neglected in this paper. Therefore, in order to give better insights into the immunological features of the non classical HLA genes and to improve overall roundness and better reading, major insertions have been made in several sections, as recommended by the reviewer. The inserted part (line 261 – 270 in the revised paper) in the section “receptor interaction of HLA-G” describes the major direct and indirect immune inhibitory mechanisms of HLA-G when bound to the receptors LILRB1 and -B2.  In detail, we tried to describe the inhibitory effects on immune cells e.g.  suppression of proliferation in CD4+ T cells mediated by HLA-G when bound to its receptors. The inserted sentences (line 288-293 of the revised paper) complete the section about HLA-G and its receptor interactions. CD160 interaction has been inserted, since this receptor interaction was missing. Also some additions have been made (line 319-324 and line 336-342) in the section “significance of HLA-E to -G for implantation/pregnancy” for better overall impression of the paper and to highlight the special immunological characteristics of implantation and pregnancy in regards to HLA-E, -F and -G. Some aspects which have been inserted in the part “significance of HLA-E to -G for implantation/pregnancy” have been readdressed in the section “HLA-G expression in cancer” (line 417-419 and line 424-431) to show the immunological similarities between pregnancy and tumor immune evasion. Regarding the immunological aspects of receptor interactions, we also inserted a section (line 211 to 215 in the revised paper) in the “Receptor interaction of HLA-E” part, describing the immune modulatory effect of HLA-E when bound to its receptors. We also did some corrections within the part “receptor interaction of HLA-E”. We inserted “B and C”, since HLA-E does also interact with the receptor NKG2B and C.  We also did some small corrections within the receptor interaction part of HLA-F. In Line 232 to 233 we added the receptors KIR2DS4 and KIR3DL2, since HLA-F is known to also interact with these KI-receptors. In Line 244 the sentence has been in inserted in order to close the section.

We also prepared a table which summarizes the cellular targets, effects and receptors of the non-classical HLA groups, as recommended by the reviewer. The tables are submitted separately.

Round  2

Reviewer 2 Report

Please correct the following:

Line 57. Remove one of the two full-stops
Line 284, remove one of the two commas
Line 442, remove one of the two commas
Line 462 please correct notiversely
Line 550 please substitute immuneoncological for immuno-oncological
Please revise syntactic structure of phase in line 549 (remove less)
Remove comma in line 572
Line 574, comma after In line with this.
Line 595, revise syntax. Furthermore, vaccination strategies….surface, have been already filed.

Author Response

We thank the reviewer for his helpful comments.

“Line 57. Remove one of the two full-stops”

As recommended by the reviewer, we deleted one of the two full-stops in line 57 of the re-revised version of the paper.

“Line 284, remove one of the two commas”

In line 284 of the re-revised version of the paper, we removed one of the two commas as suggested by the reviewer.

“Line 442, remove one of the two commas”

We deleted one of the two commas in line 443 of the re-revised paper, as recommended by the reviewer.

“Line 462 please correct notiversely”

As suggested by the reviewer, we changed the word “notiversely” to “not inversely” in line 463 of the re-revised paper.

“Line 550 please substitute immuneoncological for immuno-oncological

Please revise syntactic structure of phase in line 549 (remove less)”

In line 549 of the re-revised paper, we removed the word “less”, since the word does not contribute to the syntactic structure of the sentence. In addition, we changed the word “immuneoncological” to “immune-oncological” in line 551, as recommended by the reviewer.

 “Remove comma in line 572”

In line 574 of the re-revised version of the paper, we removed a comma after “…being”, as recommended by the reviewer.

“Line 574, comma after In line with this.”

In line 576 of the re-revised paper, we inserted a comma after the phrase “In line with this”, as suggested by the reviewer.

“Line 595, revise syntax. Furthermore, vaccination strategies….surface, have been already filed.”

We rearranged the sentence in line 597 to 602 from “Furthermore is a vaccination/immunization approaches using non-viable tumor cells or their membranes if expressing neo-antigens blocking or eradication of non-classical -HLA groups on the surface have been filed (patent pending)” to “Furthermore vaccination/immunization approaches by injecting non-viable tumor cells or their membranes (if expressing neo-antigens) could be a further therapeutical concept. The strategy of this approach is the blocking or eradication of non-classical HLA groups on the surface of the non-viable tumor cells prior to injection (patent pending) to enable recognition and antibody generation against the exposed neo-antigens.” due to false syntax.